

# Analysis of relative abundances with zeros on environmental gradients: a multinomial regression model

Fiona Chong[1,2] and Matthew Spencer[2]

[1] Systems Ecology and Resource Management Research Unit, Université Libre de Bruxelles, Bruxelles, Belgium
[2] School of Environmental Sciences, University of Liverpool, Liverpool, United Kingdom

## ABSTRACT

Ecologists often analyze relative abundances, which are an example of compositional data. However, they have made surprisingly little use of recent advances in the field of compositional data analysis. Compositions form a vector space in which addition and scalar multiplication are replaced by operations known as perturbation and powering. This algebraic structure makes it easy to understand how relative abundances change along environmental gradients. We illustrate this with an analysis of changes in hard-substrate marine communities along a depth gradient. We fit a quadratic multivariate regression model with multinomial observations to point count data obtained from video transects. As well as being an appropriate observation model in this case, the multinomial deals with the problem of zeros, which often makes compositional data analysis difficult. We show how the algebra of compositions can be used to understand patterns in dissimilarity. We use the calculus of simplex-valued functions to estimate rates of change, and to summarize the structure of the community over a vertical slice. We discuss the benefits of the compositional approach in the interpretation and visualization of relative abundance data.

## INTRODUCTION

Ecologists often analyze relative abundance data. These are sets of non-negative numbers with a fixed sum (typically 1 or 100), and are examples of compositional data, defined as equivalence classes of proportional vectors with positive components (*Pawlowsky-Glahn, Egozcue & Tolosana-Delgado, 2015*, p. 9). Compositional data present some special challenges, arising from their constrained multivariate nature, including the absence of an interpretable covariance structure and the inappropriateness of simple parametric models (*Aitchison, 1986*, chapter 3). Many of these challenges have been addressed in the last few decades, leading to a coherent set of principles for the analysis of compositional data (*Pawlowsky-Glahn & Buccianti, 2011*). Some important work on the principals of compositional data analysis was ecological. For example, *Mosimann (1962)* and *Martin & Mosimann (1965)* discussed how the nature of compositional data affects the interpretation of correlations between relative abundances of pollen types,

Corresponding author
Matthew Spencer,
m.spencer@liverpool.ac.uk

and *Billheimer, Guttorp & Fagan (2001)* developed compositional algebra as a way of studying the effects of vegetation disturbance and predator manipulation on relative abundances of arthropods. However, ecologists have made surprisingly little use of recent advances in the field. For example, *Legendre & Legendre (2012)*, one of the most important textbooks on analysis of community ecological data, does not cite any papers on compositional data analysis. Exceptions include *Jackson (1997)*, who explained how the interpretation of correlation, ordination and cluster analysis is affected by the properties of relative abundance data, *López-Flores et al. (2014)*, who showed that redundancy analysis of phytoplankton relative abundances was more ecologically informative under a compositional data analysis approach than under the usual approach, *Gross & Edmunds (2015)*, who used compositional data analysis to develop time series models for coral reef composition, and *Yuan et al. (2016)*, who used the principles of compositional data analyses in comparisons between measures of temporal change in relative abundances.

The key principle in compositional data analysis is scale invariance (*Aitchison, 1992*). This means that if **x** is a set of abundances, then $a$**x** is equivalent to **x**, for any positive real number $a$. To an ecologist, this means treating two communities as equivalent if they have the same relative abundances, even if they have different total abundances. It is straightforward to show, using the scale invariance principle, that any meaningful function of a composition can be expressed in terms of ratios of relative abundances (*Aitchison, 1992*). In addition, in most situations, subcompositional coherence is important. Suppose that two scientists are studying the same community, but one measures the abundances of all taxa, while the other measures the abundances of only some taxa. Subcompositional coherence is the requirement that their results should agree for the subset of taxa measured by both (*Aitchison, 1992*). Ecologists should care about subcompositional coherence because they are almost always studying only a subset of the taxa present in a community. For example, rare taxa may not be detected, and even if detected, it is common practice to exclude them, because modelling of patterns in abundance for such taxa is difficult (e.g., the mite data in *Borcard, Legendre & Drapeau, 1992*). Subcompositional coherence guarantees that the conclusions of an analysis of common taxa would not be changed by the addition of rare taxa. These seemingly obvious principles lead to a coherent method of manipulating relative abundance data.

For vectors representing abundances, the usual operations of addition and scalar multiplication have obvious biological meanings. However, these operations do not make sense for compositions. Instead (Section S1), there are analogous operations known as perturbation ($\oplus$) and powering ($\odot$) respectively (*Aitchison, 1986*, pp. 42, 120). Compositions with these operations form an algebraic structure known as a real vector space (*Fraleigh & Beauregard, 1995*, section 3.1). In this structure, under one of two additional conditions, there is a unique definition of the compositional difference $\ominus$ in terms of the ratios of relative abundances of corresponding taxa (*Aitchison, 1992*). The first and most important condition for ecology is that the compositional difference must not depend on changes of units for individual components, or equivalently, must not change if detection probabilities differ among taxa. The second is that the $i$th component of the transformation from one composition to another must depend only on the $i$th component

of the compositions. This is desirable because we would like to identify components of change in relative abundances associated with particular taxa. Adoption of either of these conditions leads immediately to the idea that any measure of dissimilarity between two relative abundance vectors must be perturbation invariant, i.e., it must depend only on the compositional difference between them (*Yuan et al., 2016*).

A common approach to studying variation among communities is to compute some measure $d$ of dissimilarity between pairs of communities, and then carry out graphical or numerical analyses of the resulting distance matrix (*Legendre & Legendre, 2012*, chapter 7). This has the potential to mislead if the measure of dissimilarity is not perturbation invariant (Section S2). Most of the popular measures of community dissimilarity are not perturbation invariant (*Spencer, 2015*, Appendix B). In contrast, the Aitchison distance (*Aitchison, 1992*) is a well-established perturbation-invariant measure of dissimilarity between compositions. Thus, analyses of dissimilarity between relative abundances should be based on the Aitchison distance, rather than on currently-popular measures of community dissimilarity. The value of the Aitchison distance is now recognized in microbiome analysis (*Gloor et al., 2017*), but it remains little used in most areas of ecology.

Model-based analysis is an increasingly popular alternative way of analyzing differences between communities (*Warton et al., 2015*). Model-based methods allow appropriate modelling of the observation process, which often leads to mean–variance relationships different from those implicit in widely-used measures of dissimilarity (*Warton, Wright & Wang, 2012*). Model-based methods are generally more flexible, interpretable and efficient than dissimilarity-based methods (*Warton et al., 2015*). For example, once a parametric model has been fitted to a set of communities along an environmental gradient, the function that describes expected values can be differentiated to find the rate of change of the community along the gradient, and integration can be used to find the mean community over the entire gradient. Even when dissimilarities are directly of interest, a parametric model is useful in understanding how expected dissimilarity depends on distance along the gradient. However, an overlooked distinction between model-based and dissimilarity-based methods is that most model-based methods (e.g. *Wang et al., 2012*) are designed for abundance data, while most dissimilarities are designed for relative abundance data. Communities are often treated as equivalent if they have the same ''shape'' (i.e., if they represent equivalent compositions, in the language of compositional data analysis) regardless of differences in ''size'' (total abundance). Failing to recognize this distinction can lead to misinterpretation of the results of common analyses such as permutation-based anova (*Greenacre, 2017*). Also, in some cases (e.g., point counts from vegetation and on coral reefs, pollen counts, and environmental sequencing data), only relative abundances are available. Thus, there is a need for model-based analyses of relative abundance data. It seems likely that compositional data analysis, combined with the calculus of simplex-valued functions (*Egozcue, Jarauta-Bragulat & Díaz-Barrero, 2011*; *Pawlowsky-Glahn, Egozcue & Tolosana-Delgado, 2015*, chapter 9), will meet this need.

Here, we show how the vector space structure of the simplex provides a coherent way to study changes in community composition along environmental gradients. We show that a low-order polynomial provides a good model for the composition of a community

of sessile hard-substrate marine organisms over a depth gradient. We illustrate the use of Aitchison distance as a principled measure of dissimilarity between communities, and use the algebraic structure of the simplex to understand how dissimilarity depends on depth. In particular, we determine the conditions for the same community composition to occur at different depths. We use the calculus of simplex-valued functions to answer two biological questions: at what depth is the community changing fastest, and which taxa dominate the mean composition over the entire depth range?

## MATERIALS AND METHODS

### Location

We studied the community of sessile hard-substrate marine organisms on the walls of Salthouse Dock (53.4006°N, 2.9898°W), Port of Liverpool, United Kingdom. Salthouse Dock is part of the southern dock system on the River Mersey (Fig. S1), connected to Wapping Dock to the South, Albert Dock to the West and Canning Dock to the North via Albert Dock. The docks fell into disuse in the 1970s, but were dredged and reopened for recreational use in 1981 (*Fielding, 1997*, pp. 10–14). Since then, they have been redeveloped as part of a commercial project, and with the completion of the Liverpool Canal Link, are now also connected to the Leeds-Liverpool Canal (*Coutts, Pellizzon & Alderdice, 2012*). The regenerated docks are a shallow, semi-enclosed brackish water habitat, with salinity between 22‰ and 33‰ in the South Docks (*Fielding, 1997*, pp. 17, 70).

### Video transects

An OpenROV v2.8 remotely-operated vehicle (OpenROV, Berkeley, CA, USA) with an IMU/Depth sensor and the Pro Camera-HD Upgrade (1,080 p) was used to take 31 approximately vertical transects from surface to bottom, haphazardly spaced along the northern and eastern walls of Salthouse Dock, on 2 February 2017 (Fig. S1, inset). The distance from the wall was typically around 0.3 m to 0.4 m, giving a field of view with an area of approximately 0.29 m$^2$ to 0.51 m$^2$. The field of view was not known exactly because the lasers on the ROV, intended to indicate a known distance on the images, malfunctioned. However, the field of view was always large enough to contain many organisms, so that the relative abundances are unlikely to depend on the exact area sampled. In addition, as described below, we included a random intercept term in the model, which will capture some of the effects of variation in field of view. A GoPro HERO3+ Black Edition (GoPro, San Mateo, CA, USA) was also attached to the ROV to provide an extra source of footage with higher resolution but more distortion. The OpenROV videos and telemetry data were recorded in the inbuilt Cockpit software (v30.1.0 with software patch release). The video and data files were downloaded and python scripts were written to overlay depth data on the corresponding videos.

### Image analysis

Four still images were captured per transect video at varying depths from 0.11 m to 3.72 m (except one transect where five stills were taken), making 125 still images in total. These stills were selected by viewing the video frame by frame, based on the clarity of the image,

which is generally best when the ROV is at an optimum distance from the wall and moving relatively slowly. On each image, the taxon present at each of 100 randomly-selected points was identified by human visual curation and recorded using the JMicroVision v1.2.7 image analysis software (*Roduit, 2008*, Fig. 1). The process of extracting data from video transects is summarized in Fig. 2. Where necessary, further viewing of surrounding frames from the ROV video and supplementary GoPro footage were also used in identification. Most identifications (Table 1) were confirmed using specimens collected from near the surface, following *Hayward & Ryland (1995)*. For the non-native colonial sea squirt *Botrylloides violaceus*, we used the Marine Life Information Network (*Snowden, 2008*). Where an organism was growing on top of another, the organism taking up space on the wall was recorded. If positive identification was not possible, the point was skipped and another point drawn. "Bare wall" was recorded if no macroscopic organism was present, or (as often occurred near the bottom) if the wall was covered by grey detritus, so that any macroscopic organisms which may have been present were not visible. Point counts were exported from JMicroVision into ASCII text files, which were combined using an R 3.4.0 script (*R Core Team, 2017*) into a single file with depth data.

## Data analysis
### Data aggregation
Due to the rarity of barnacles and *Stomphia coccinea* (three and one individuals respectively), these two taxa were excluded from the analysis. Points where these taxa were sampled were not redrawn, leaving one still with 91 points, three with 99, and the remainder with 100 points. The remaining taxa were combined into eight categories, consisting of organisms that were ecologically similar and/or could not be reliably distinguished: algae (red and green), *Aurelia aurita* polyps, *Bugula spp.*, colonial ascidians (*Botryllus schlosseri*, *Botrylloides leachii* and *Botrylloides violaceus*), *Diadumene cincta*, solitary ascidians (*Ciona intestinalis* and *Styela clava*), sponges (*Halichondria spp.* and others), *Mytilus edulis*. We also included the "bare wall" category (for the absence of macroscopic organisms, although usually there was a biofilm of microscopic algae and bacteria, or a layer of detritus).

### Statistical model
Let the counts in the $i$th observation (still image) be $\mathbf{y}_i = (y_{i,1}, y_{i,2}, \ldots, y_{i,9})^T$, where $y_{i,j}$ is the observed count of the $j$th taxon in the $i$th observation, and let $n_i = \sum_{j=1}^{9} y_{ij}$ be the total number of points counted for the $i$th observation (usually 100 in our data). Our model is

$$\mathbf{y}_i \sim \text{multinomial}(n_i, \boldsymbol{\rho}_i),$$
$$\boldsymbol{\rho}_i = \text{ilr}^{-1}\mathbf{x}_i,$$
$$\mathbf{x}_i = \boldsymbol{\beta}_0 + \boldsymbol{\beta}_1 z_i + \boldsymbol{\beta}_2 z_i^2 + \boldsymbol{\varepsilon}_i, \tag{1}$$
$$\boldsymbol{\varepsilon}_i \sim N(\mathbf{0}, \boldsymbol{\Sigma}).$$

In a non-Bayesian context, this model would be referred to as a multivariate generalized linear mixed model (*Agresti, 2002*, p.492), with a multinomial response distribution, an isometric logratio (ilr: *Egozcue et al., 2003*) link function, linear predictor $\mathbf{x}_i$ and random effects $\boldsymbol{\varepsilon}_i$. The vector $\boldsymbol{\rho}_i$ is the expected relative abundance of each taxon. The multinomial

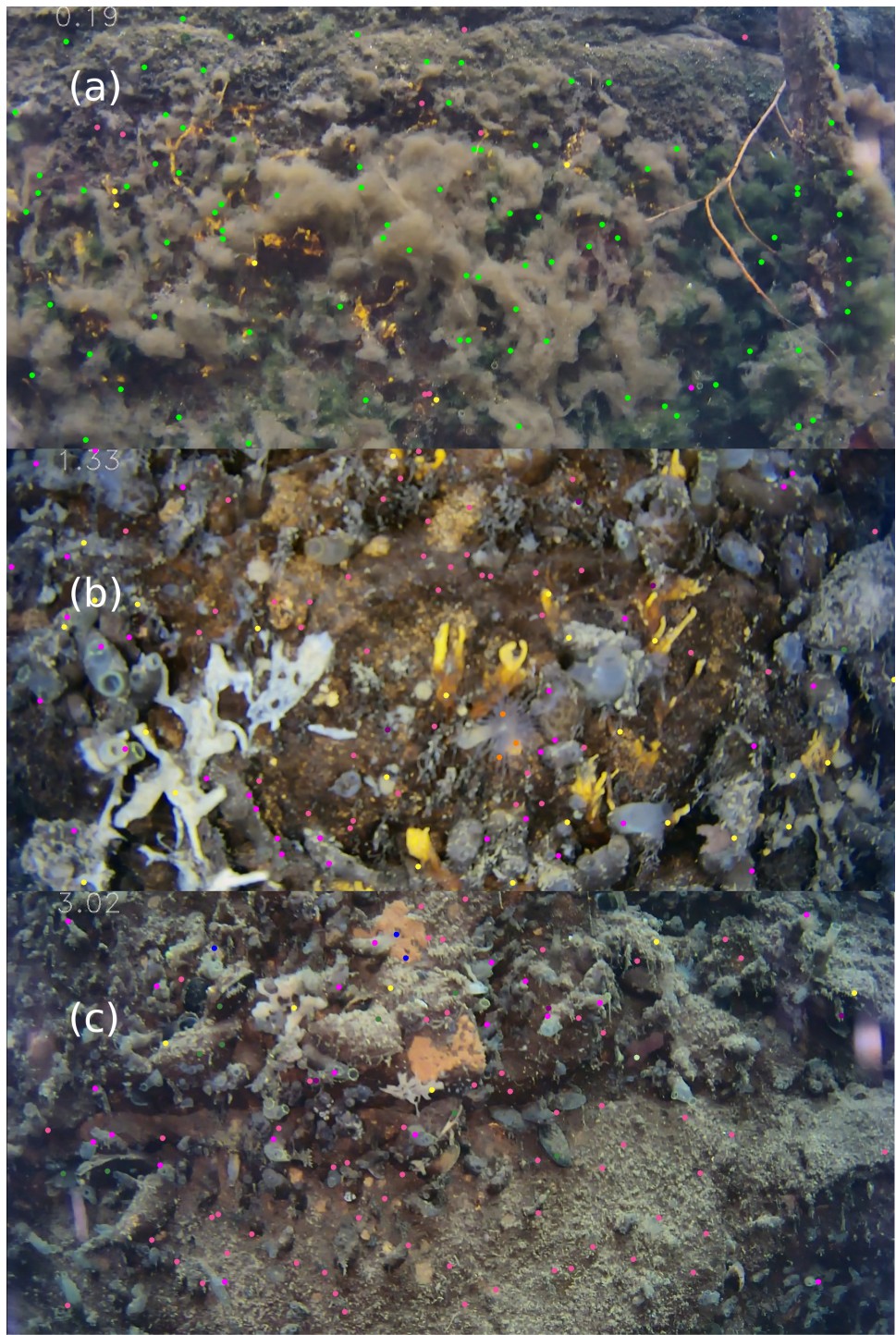

**Figure 1** **ROV still images from (A) 0.19 m, (B) 1.33 m and (C) 3.02 m, with 100 point counts each.**
Bright green dots correspond to green algae, pink dots to bare wall, violet to *Ciona intestinalis*, yellow to
*Halichondria spp.*, purple to *Bugula spp.*, orange to *Diadumene cincta*, green to *Mytilus edulis*, blue to other
sponges and off-white to *Botrylloides violaceus*. Photos: Fiona Chong.

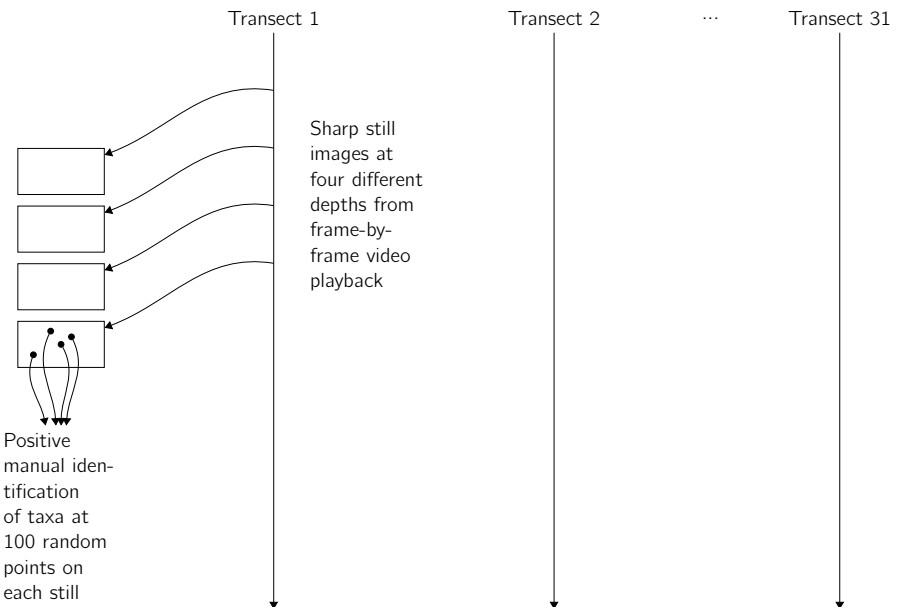

**Figure 2** Summary of the process by which count data were extracted from video transects.

observation model arises from the assumption that individual points within a still are drawn independently from a categorical distribution with probabilities $\boldsymbol{\rho}_i$ (*Johnson, Kotz & Balakrishnan, 1997*, p. 33). The ilr link function transforms the 8-simplex into an unconstrained 8-dimensional real space, with an ilr coordinate system described below. The linear predictor $\mathbf{x}_i$ is an 8-dimensional vector in ilr coordinates, and depends on $\boldsymbol{\beta}_0$, $\boldsymbol{\beta}_1$ and $\boldsymbol{\beta}_2$, the unknown 8-dimensional intercept and linear and quadratic depth coefficient vectors respectively, and on $z_i$, the centred and scaled depth for the $i$th observation. The observation-specific intercepts $\boldsymbol{\varepsilon}_i$ are drawn from an 8-dimensional multivariate normal distribution in ilr coordinates, with mean vector $\mathbf{0}$ and covariance matrix $\Sigma$. These intercepts deal with extra-multinomial variation (overdispersion) arising from factors such as clustering due to the spatial extension of organisms and unmeasured covariates (*McCullagh & Nelder, 1989*, pp. 124–125, 174). In particular, in our data, variation in the distance of the ROV from the wall is likely to lead to varying amounts of overdispersion among stills. This treatment of overdispersion leads to a normal distribution of expected values on the simplex, in the sense of *Pawlowsky-Glahn, Egozcue & Tolosana-Delgado (2015*, p. 114). This distribution is much more flexible than, for example, a Dirichlet distribution, although there are many other reasonable choices.

It is important that observations $\mathbf{y}_i$ including zero counts are in the support of the multinomial distribution, and that fitting the model involves back-transforming the linear predictor (which is always in the domain of $\text{ilr}^{-1}$), not an ilr transformation of $\mathbf{y}_i$. Thus, no special treatment of zeros (such as pseudocounts) is necessary. We fitted this model using Bayesian estimation via the NUTS algorithm (*Hoffman & Gelman, 2014*). NUTS is derived from Hamiltonian Monte Carlo, in which the problem of sampling from the posterior

**Table 1  List of species identified from stills and samples.**

*Aurelia aurita*

*Botryllus schlosseri*

*Botrylloides leachii*

*Botrylloides violaceus*

*Bugula spp.*

*Ciona intestinalis*

*Diadumene cincta* (some individuals may be *Metridium senile* (*Neal, 2007*))

Green algae

*Halichondria spp.*

*Mytilus edulis*

Other sponges

Red algae

*Stomphia coccinea*

*Styela clava*

Unidentified barnacle

distribution of interest is formulated in terms of simulating the dynamics of a physical system with position, potential energy and momentum (*Neal, 2011*). This can explore the state space much more rapidly than random-walk methods such as the Metropolis–Hastings algorithm. NUTS improves on Hamiltonian Monte Carlo by requiring much less fine-tuning, and is implemented in the Stan programming language (*Carpenter et al., 2017*). We give more details in Section S3. We checked the model's performance using a simulation study (Section S4). We used a Bayesian approach, despite the additional computation it involves, because it leads almost automatically to estimates of uncertainty in the compositional analyses described below. We compared the performance of this model against models with only a linear depth effect and with a cubic depth effect, using leave-one-out cross-validation to estimate the expected log predictive density for a new data set (Section S5).

The vector $\boldsymbol{\rho}_i$ consists of non-negative elements with a fixed sum of 1, and is therefore a composition. The sum constraint, and associated constraints on the covariance structure of compositions, make it difficult and inconvenient to specify sufficiently flexible parametric models for untransformed compositions (*Aitchison, 1986*, chapter 3). The most popular modern approach to analysis of compositional data is to transform an $s$-part composition into an unconstrained real space with $s - 1$ dimensions. We chose an isometric logratio transformation (*Egozcue et al., 2003*), which is an isomorphism (so that perturbation and powering in the simplex correspond to ordinary vector addition and scalar multiplication in the real space) and an isometry (so that distances under an appropriate norm in the simplex correspond to Euclidean distances in the real space).

The coordinates in an ilr coordinate system represent logcontrasts between groups of taxa (loglinear combinations of relative abundances whose coefficients sum to zero: *Aitchison, 1986*, p. 84). The ilr transformation is defined by a basis matrix, constructed from a set of $s - 1$ orthogonal logcontrasts. In principle, such logcontrasts can be very informative

biologically. For example, in our study we would expect the logcontrast between algae and animals to decrease with depth, because algae were the only photosynthetic organisms included. We would expect the logcontrast between predatory and nonpredatory animals to increase with depth, because predatory animals do not rely on photosynthetic food, and we would expect the logcontrast between the two predators, *A. aurita* and *D. cincta*, to increase with depth because *A. aurita* polyps have a strong preference for dark locations (*Ishii & Shioi, 2003*).

In order to fit the model, we used the isometric logratio transformation with the default basis matrix in the R package 'compositions', version 1.40-1 (*Van den Boogaart & Tolosana-Delgado, 2008*). Our results do not depend on this choice of basis, but if it is important to be able to interpret logratio coordinates, an appropriate basis can be chosen by sequential binary partition (*Egozcue & Pawlowsky-Glahn, 2005*). We describe such a basis in the Section S6. Meaningful bases can also be constructed from hierarchical clustering of environmental preferences (*Morton et al., 2017*) or from a phylogeny (*Silverman et al., 2017*). Advantages and disadvantages of the ilr transformation, compared to other transformations, are discussed in *Bacon-Shone (2011*, section 1.5).

Because the isometric logratio transformation is an isomorphism between the simplex with Aitchison geometry (*Pawlowsky-Glahn & Egozcue, 2001*) and the ordinary real space, we can back-transform the deterministic part of Eq. (1) to obtain an expression in terms of perturbation and powering in the simplex:

$$M(\boldsymbol{\rho}_i) = \mathrm{ilr}^{-1}\left(\boldsymbol{\beta}_0 + \boldsymbol{\beta}_1 z_i + \boldsymbol{\beta}_2 z_i^2\right)$$
$$= \boldsymbol{\gamma}_0 \oplus (z_i \odot \boldsymbol{\gamma}_1) \oplus (z_i^2 \odot \boldsymbol{\gamma}_2),$$

where $\boldsymbol{\gamma}_j = \mathrm{ilr}^{-1}(\boldsymbol{\beta}_j)$, $j = 0, 1, 2$. The composition $M(\boldsymbol{\rho}_i)$ is the metric centre (*Pawlowsky-Glahn & Egozcue, 2001*) of the distribution of $\boldsymbol{\rho}_i$, an appropriate measure of location for compositions (*Aitchison, 1989*).

To make the behaviour of the predictions for rare taxa more obvious, we also examined the predictions on a centred logratio (clr) scale, in which the value on the $y$-axis is the log of the ratio of the corresponding component to the geometric mean of all components (*Aitchison, 1986*, p. 79). A constant slope on the clr scale corresponds to constant proportional change in the relative abundance of a given taxon. This is also true of the ilr scale, but not of the original proportions. We use the clr scale here because, unlike the ilr scale, it has one coordinate associated with each taxon. For the same reason, clr coordinates are usually chosen as rays in a compositional biplot (*Aitchison & Greenacre, 2002*). However, it is important to remember that slopes on the clr scale are dependent on the set of taxa analyzed. In addition, although there are $s$ clr coordinates, points in the clr space are constrained to lie in an $(s-1)$-dimensional hyperplane in which the sum of the coordinates is zero. This means, that, for example, covariance matrices in the clr scale are singular (*Aitchison, 1986*, pp. 78–81).

### Comparison with non-metric multidimensional scaling

We contrasted our approach with what is likely to be the most popular alternative in marine ecology, a non-metric multidimensional scaling of the raw counts. We used the metaMDS() function in R package 'vegan', with default options (square root transformation, Wisconsin double standardization, Bray–Curtisrtis dissimilarity). For comparison, we plotted the first two principal components of the posterior mean still-specific predictions in ilr coordinates.

### Alternative models

We also considered multinomial regression fitted by penalized likelihood using 'glmnet' (*Friedman, Hastie & Tibshirani, 2010*), and two naive models that are easy to fit: overdispersed Poisson regression using HMSC (*Ovaskainen et al., 2017*), which does not respect the multinomial sums, and multivariate linear regression on ilr-transformed counts with the addition of three different kinds of pseudocount (*Martín-Fernandez, Palarea-Albaladejo & Olea, 2011*). For details, see Section S7.

### Community dissimilarity

As described above, most of the common measures of dissimilarity between communities are not perturbation invariant. In the Aitchison geometry, the obvious perturbation invariant measure of difference between two $s$-part compositions is the Aitchison distance (the Aitchison norm of the compositional difference), defined by

$$d_a(\boldsymbol{\rho}_1, \boldsymbol{\rho}_2) = \|\boldsymbol{\rho}_1 \ominus \boldsymbol{\rho}_2\|_a$$

$$= \left[\sum_{i=1}^{s}\left(\log\frac{\rho_{1,i}}{g(\boldsymbol{\rho}_1)} - \log\frac{\rho_{2,i}}{g(\boldsymbol{\rho}_2)}\right)^2\right]^{1/2}$$

$$= \left[\sum_{j=1}^{s-1}(x_{1,j} - x_{2,j})^2\right]^{1/2}$$

(*Aitchison, 1992*; *Egozcue et al., 2003*), where $g(\boldsymbol{\rho}_k)$ denotes the geometric mean of the parts of a composition, and $x_{k,j}$ denotes the $j$th ilr coordinate of $\mathbf{x}_k = \mathrm{ilr}(\boldsymbol{\rho}_k)$, $k = 1, 2$. The last line gives the Aitchison distance as the Euclidean distance in ilr coordinates (*Egozcue et al., 2003*). It is immediately obvious that the Aitchison distance is perturbation invariant, because $(\mathbf{a} \oplus \boldsymbol{\rho}_1) \ominus (\mathbf{a} \oplus \boldsymbol{\rho}_2) = \boldsymbol{\rho}_1 \ominus \boldsymbol{\rho}_2$, by the associative, commutative and identity properties of the vector space. Under this approach, the dissimilarity between the expected compositions $\boldsymbol{\rho}_1, \boldsymbol{\rho}_2$ is given by

$$\|\boldsymbol{\rho}_1 \ominus \boldsymbol{\rho}_2\|_a = \|[\boldsymbol{\gamma}_0 \oplus (z_1 \odot \boldsymbol{\gamma}_1) \oplus (z_1^2 \odot \boldsymbol{\gamma}_2)] \ominus [\boldsymbol{\gamma}_0 \oplus (z_2 \odot \boldsymbol{\gamma}_1) \oplus (z_2^2 \odot \boldsymbol{\gamma}_2)]\|_a$$

$$= |z_1 - z_2| \|\boldsymbol{\gamma}_1 \oplus [(z_1 + z_2) \odot \boldsymbol{\gamma}_2]\|_a, \tag{2}$$

using the identity, commutative, associative and distributive properties of the vector space to simplify.

The Aitchison distance has a biological meaning in terms of population growth. In temporal comparisons, the Aitchison distance between two sets of relative abundances is proportional to the among-taxon standard deviation of proportional population growth

rates (*Spencer, 2015*). In spatial comparisons, we can therefore think of the Aitchison distance as measuring the among-taxon variability in proportional population growth rates that is needed to transform one set of relative abundances into another, over a given time interval. This property is important because in a closed system, population growth is the only way to transform one set of relative abundances into another. No other measure of community dissimilarity has this interpretation.

The simplex with Aitchison geometry is a normed vector space (*Egozcue et al., 2003*). Thus $|\boldsymbol{\rho}_1 \ominus \boldsymbol{\rho}_2|_a = 0$ if and only if $\boldsymbol{\rho}_1 \ominus \boldsymbol{\rho}_2 = \mathbf{0}$, where $\mathbf{0}$ is the identity element in the simplex (e.g. *Horn & Johnson, 1985*, p. 259). From Eq. (2), assuming that $\boldsymbol{\gamma}_1 \neq \mathbf{0}$ and $\boldsymbol{\gamma}_2 \neq \mathbf{0}$, this happens when either $z_1 = z_2$ (the two compositions are at the same depth) or $\boldsymbol{\gamma}_2 = \left(-\frac{1}{z_1+z_2}\right) \odot \boldsymbol{\gamma}_1$ (the coefficient of squared depth is a powering of the coefficient of depth). Thus, if we plot dissimilarity on a grid of depths, there will always be zeros on the main diagonal, because communities at the same depth have the same expected composition. There may also be communities at different depths with the same expected composition, along a counter-diagonal where centred and scaled depth has a constant sum, but only in the special case where $\boldsymbol{\gamma}_2$ is a powering of $\boldsymbol{\gamma}_1$ (or equivalently, where $\boldsymbol{\beta}_2$ is a scalar multiple of $\boldsymbol{\beta}_1$ in ilr coordinates).

We calculated posterior distributions of dissimilarities among 100 equally-spaced expected compositions between the minimum and maximum depths, both including and excluding bare wall. We plotted the posterior mean dissimilarity matrix, and the widths of the 95% highest posterior density intervals. We only report the results including bare wall here, because those excluding bare wall were very similar. Note that it is valid to exclude some parts of the composition if necessary, because the subcompositional coherence property means that such exclusion will not affect relationships among the remaining parts (*Aitchison, 1994*).

### Rate of change of community composition with depth

The community is changing rapidly with respect to depth if a small increase in depth leads to a large difference in composition. In order to correctly evaluate this change, we need an appropriate definition of difference in composition. Given the geometry of the simplex, the difference in composition between depths $z$ and $z + h$ is naturally expressed as $\mathbf{f}(z + h) \ominus \mathbf{f}(z)$. Then letting $h$ go to zero leads to the obvious definition of the derivative $D^{\oplus}\mathbf{f}$ of a simplex-valued function $\mathbf{f}$,

$$D^{\oplus}\mathbf{f}(z) = \lim_{h \to 0}\left(\frac{1}{h} \odot (\mathbf{f}(z + h) \ominus \mathbf{f}(z))\right),$$

provided this limit exists (*Egozcue, Jarauta-Bragulat & Díaz-Barrero, 2011*, section 12.2.2). Using the rules for differentiation of simplex-valued functions (*Egozcue, Jarauta-Bragulat & Díaz-Barrero, 2011*, section 12.2.2), in our model, the derivative of expected community composition $M$ with respect to depth, at a depth of $z$, is

$$D^{\oplus}M(z) = \boldsymbol{\gamma}_1 \oplus (2z \odot \boldsymbol{\gamma}_2).$$

This is itself a composition. If we want a scalar measure of rate of change, the obvious choice is the norm of this derivative. It is intuitively obvious that the usual Euclidean norm is not

appropriate, because the zero element for compositions (with all parts equal, corresponding to no change in composition with respect to depth) does not have zero Euclidean norm. Instead, we use the Aitchison norm $|D^{\oplus}M(z)|_a$ (*Egozcue et al., 2003*), which is zero in the situation where there is no change in composition with respect to depth, and is used in the definition of a limit in the simplex (*Egozcue, Jarauta-Bragulat & Díaz-Barrero, 2011*, Definition 12.2.1). The easiest way to think of this norm is that it is equal to the Euclidean norm of the derivative in isometric logratio coordinates. We evaluated the posterior distribution of this scalar measure of rate of change at 100 equally-spaced depths over the observed depth range.

It is important to remember that we are measuring proportional change: doubling of relative abundance means the same thing whether the initial relative abundance is low or high. This is an essential property, because relative abundances have meaning only in relative terms. In addition, an increase in relative abundance of a taxon may occur in several different ways. For example, the absolute abundance of a taxon may increase while absolute abundances of other taxa remain constant, or the absolute abundance of a taxon may decrease while absolute abundances of other taxa decrease more. In compositional data analysis (and in ecological situations where the focus is on relative abundances), these situations are equivalent.

In order to show how the compositional approach leads to different results from widely-used approaches in ecology, we plotted Bray–Curtis dissimilarities between adjacent predicted compositions (on a grid of 100 equally-spaced depths) against depth (Section S8). This gives a rough estimate of the relationship between rate of change in community composition and depth, because the depth intervals are small. In order to show that this is a potentially general result, we performed a similar analysis for the mite data set of *Borcard, Legendre & Drapeau (1992)*. We fitted a compositional regression model with linear effects of substrate density and water content, with the same multinomial observation model as for the marine community data, and plotted Bray–Curtis dissimilarities between adjacent predicted compositions at equally-spaced values of each explanatory variable, with the other variable held constant (Section S9).

### Depth-integrated relative abundances

Over a vertical slice from surface to bottom, a taxon that has high relative abundance over a small range of depths may be unimportant compared to a taxon that has moderate relative abundance at all depths. We therefore want some measure of the ''mean'' relative abundances over a vertical slice. The arithmetic mean is not appropriate for compositional data. For example, with a banana-shaped distribution, the arithmetic mean may lie completely outside the cloud of observations. The metric centre is a more appropriate measure of the centre of a compositional distribution which avoids these problems (*Aitchison, 1989*). However, taking a sample estimate of the metric centre over all depths is problematic when there are zero counts. Zeros are difficult to deal with in compositional data analysis (*Martín-Fernandez, Palarea-Albaladejo & Olea, 2011*), and in this context, will lead to the estimate of the centre being undefined. In addition, if the depth distribution of samples is not uniform, the sample estimate of the centre will be biased. Thus, integrating

the model-estimated composition over the full range of depths may be a better way to summarize the structure of the community.

The mean of a real function $f$ of one variable over the interval $[a, b]$ is

$$\frac{1}{b-a} \int_a^b f(x)dx,$$

which can be thought of as the value of the constant function whose integral over $[a, b]$ is the same as that of $f$ over the same interval (*Riley, Hobson & Bence, 2002*, pp. 73–74). If we treat community composition as a simplex-valued function of depth, then the analogous mean of this function over the full range of depths gives the composition representing the relative abundance of each part over a vertical slice from top to bottom of the dock wall. Let $[S, D]$ be the depth range, from shallow to deep. Using the rules for integration of simplex-valued functions (*Egozcue, Jarauta-Bragulat & Díaz-Barrero, 2011*, section 12.3.2), the required mean value is

$$\frac{1}{D-S} \odot \left[ (z \odot \boldsymbol{\gamma}_0) \oplus \left( \frac{z^2}{2} \odot \boldsymbol{\gamma}_1 \right) \oplus \left( \frac{z^3}{3} \odot \boldsymbol{\gamma}_2 \right) \right]_S^D.$$

We evaluated the posterior distribution of this mean value.

## RESULTS

### Trends in composition with depth

Images at different depths often showed large differences in relative abundances (Fig. 1). For example, Fig. 1A, at 0.19 m, was dominated by green algae. Figure 1B, at 1.33 m, was dominated by bare wall, *Halichondria* spp. and *C. intestinalis*, and also had some *D. cincta* and *Bugula* spp. Figure 1C, at 3.02 m, still had fairly high relative abundance of *Halichondria* spp. and *C. intestinalis*, and also a moderate relative abundance of *M. edulis*. However, large areas of the lower part of this image were covered by grey detritus and were therefore assigned to bare wall.

Over all the images, there were obvious changes in the relative abundance of bare wall, *Bugula*, solitary ascidians, algae and sponges with depth (Figs. 3A–3E, circles), while the relative abundances for the rare taxa *D. cincta*, *M. edulis*, *A. aurita* and colonial ascidians had apparently weaker trends (Figs. 3F–3I, circles). However, note that, as outlined below, the relative scale on the main panels in Fig. 3 means that the strength of trends is not always easy to judge. The fitted model (Fig. 3, lines) closely tracked the pattern in the observations, indicating that a quadratic model is a plausible description of changes in relative abundance over the depth gradient (the linear model was much worse than the quadratic, and the cubic model was little different from the quadratic: Section S5 and Fig. S4). The relative abundance of bare wall increased from about 0.1 to 0.4 between 0 m and 1 m, remained fairly constant until 2 m, and increased again to about 0.9 in the deepest samples (Fig. 3A). This is a more complicated pattern than could be produced by a quadratic function in an unrestricted space. The cover of algae dropped dramatically from around 0.8 at the surface to almost nothing just after 1 m (Fig. 3C). The remaining three taxa with moderately high relative abundances at some depths (*Bugula*, solitary ascidians and sponges: Figs. 3B, 3C,

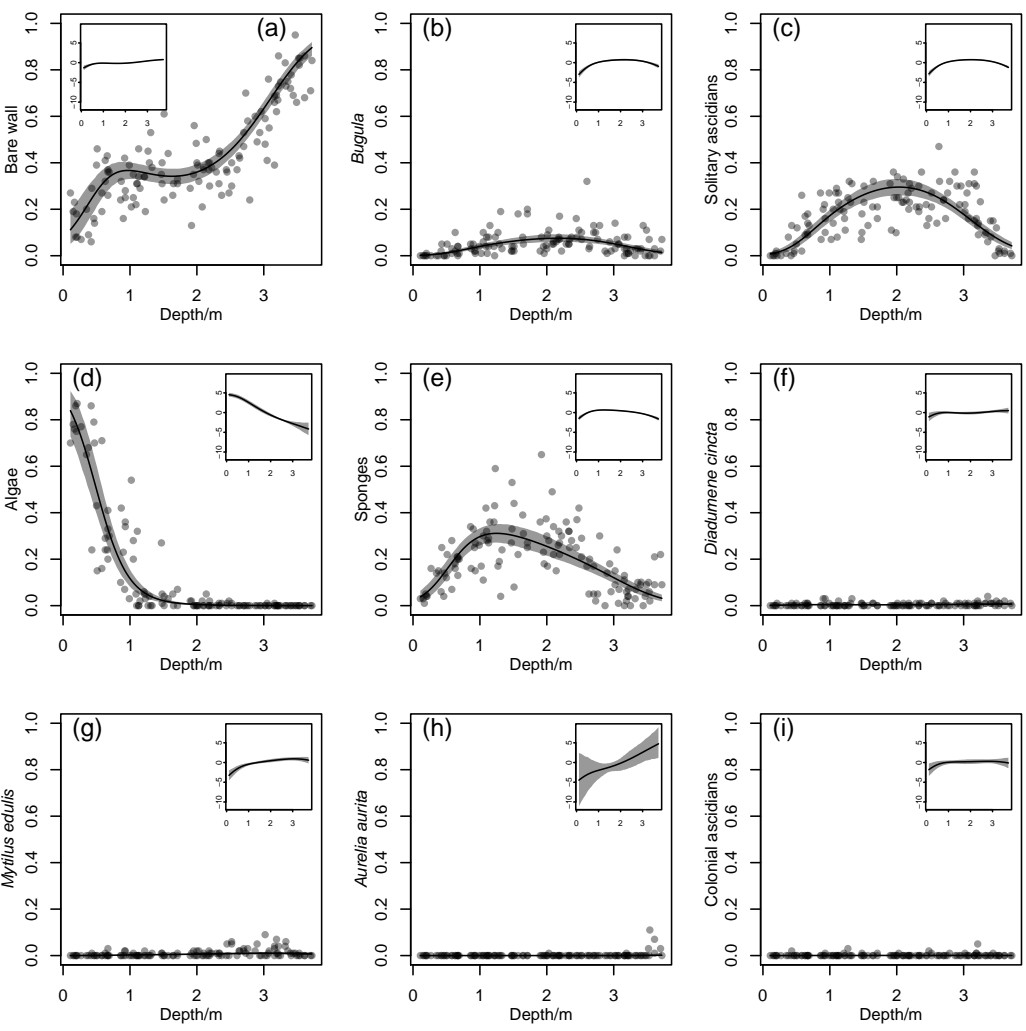

**Figure 3** **Estimated relationships between relative abundance and depth for (A) bare wall, (B)** *Bugula* **spp., (C) solitary ascidians, (D) algae, (E) sponges, (F)** *Diadumene cincta,* **(g)** *Mytilus edulis,* **(h)** *Aurelia aurita,* **(I) colonial ascidians.** Circles are sample estimates of relative abundance from point counts. Grey bands are 95% highest posterior density (HPD) credible bands, and black lines are posterior means. Insets: posterior means and 95% HPD credible bands on a centered logratio scale, in which the value on the *y*-axis is the log of the ratio of the corresponding component to the geometric mean of all components

3E) were all absent at the surface and rare in the deepest samples, with peaks at intermediate depths (around 1 m for sponges, 2 m for *Bugula* and solitary ascidians).

For the rare taxa, centred logratio plots showed that although the predicted relative abundances were low everywhere, the proportional changes in predicted relative abundance (Figs. 3F to 3I, insets) were comparable to those for common taxa. All the rare taxa had lower predicted relative abundances near the surface, with *D. cincta* (Fig. 3F) showing little change at mid depths, *M. edulis* (Fig. 3G) and colonial ascidians (Fig. 3I) decreasing in abundance in the deepest samples, and *A. aurita* (Fig. 3H) increasing steadily with depth.

Overall, the trend for *A. aurita* was potentially the strongest, but with high uncertainty. The centred logratio trends are in accordance with the observations. For example, *A. aurita* was only observed occasionally. However, when it was observed, it was below 3 m and in dense aggregations of small polyps, especially on downward-facing parts of the dock wall. The fitted trend ensures that the probability of a non-zero count is very low except for images deeper than 3 m.

Inspection of predictions in ilr coordinates with an informative basis (Fig. S6) confirmed that as expected, the logcontrast between algae and animals decreased with depth, and that the logcontrast between *A. aurita* and *D. cincta* decreased with depth. The logcontrast between predatory and filter-feeding animals increased with depth for depths greater than about 1 m, but unexpectedly decreased with depth for depths less than about 1 m.

Alternative models (Section S7 and Fig. S5) made similar prediction to those from our approach for taxa with high relative abundances. All the alternative methods other than Perks pseudocounts and 'glmnet' tended to overpredict relative abundances of rare taxa. Nevertheless, we would expect that for a moderately large, well-behaved data set such as this one, any reasonable regression approach should perform adequately.

Non-metric multidimensional scaling on the raw counts failed to reveal the effects of depth (Fig. S7A). In contrast, the depth effect was clearly visible in the first two principal components of still-specific predictions in ilr coordinates (Fig. S7B).

## Community dissimilarity

Dissimilarity between expected composition, measured as the Aitchison distance (Eq. (2)) was small for small differences in depth (Fig. 4, upper triangle, dark colours), and increased with increasing difference in depth. The uncertainty in dissimilarity behaved in a similar way (Fig. 4, lower triangle). There was no counter-diagonal pattern of similar communities at widely-separated depths, suggesting that communities at different depths never have the same expected composition. The 'Community dissimilarity' section in the Methods gives a way to check this property. We showed there that communities at different depths can only have the same expected composition if the coefficient $\gamma_2$ of squared depth in the simplex is a powering of the coefficient $\gamma_1$ of depth in the simplex. If this property holds, then the compositional line of powerings of $\gamma_1$ will pass through the composition $\gamma_2$. Figure 5 shows that for the subcomposition consisting of bare wall, algae and sponges, the high-density region of the posterior distribution of the line of powerings of $\gamma_1$ (Fig. 5, lines) does not pass through the high-density region of the posterior distribution of $\gamma_2$ (Fig. 5, points). Thus $\gamma_2$ is not likely to be a powering of $\gamma_1$, and dissimilarity is not likely to be zero for communities with a non-zero difference in depth. Although expected relative abundance may be the same at widely-separated depths for individual taxa (e.g., sponges, Fig. 3E), this pattern does not coincide across taxa.

## Rate of change of community composition with depth

The posterior mean rate of change of community composition with respect to depth was highest at the surface, decreased with increasing depth until just below 2 m, and increased again until the bottom was reached (Fig. 6, white line). Although the 95%

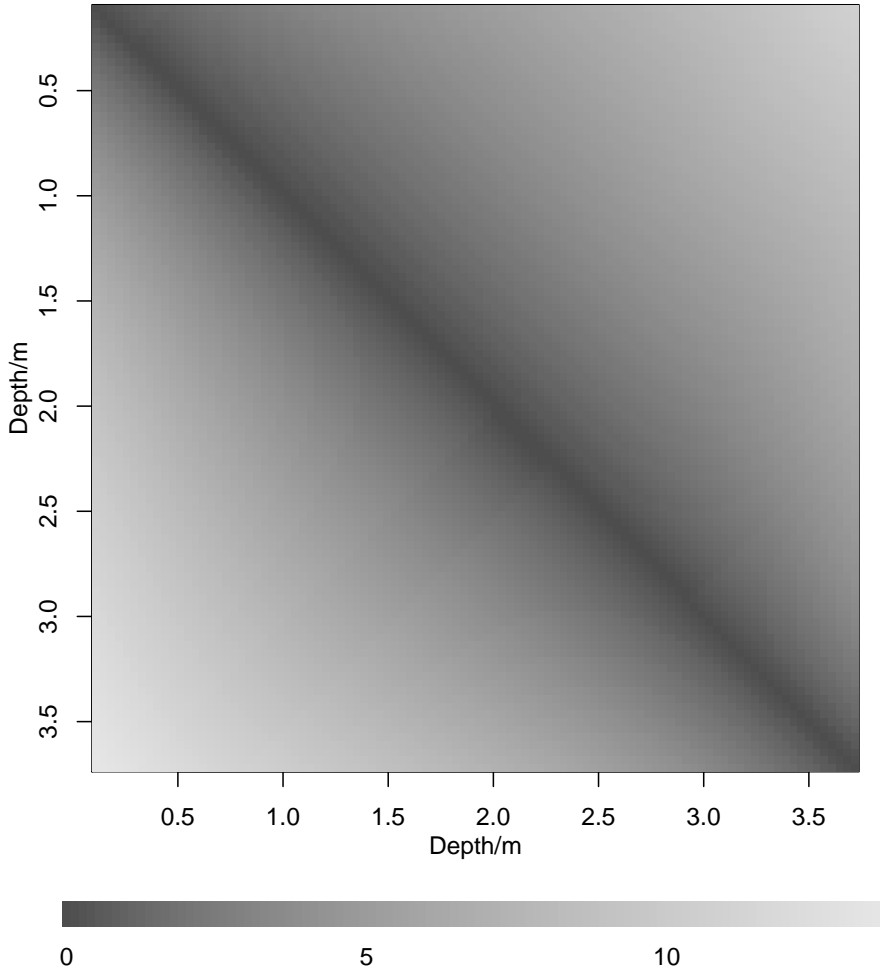

**Figure 4   Dissimilarity matrix based on Aitchison distance between expected composition at different depths.** Posterior mean (upper triangle) and width of 95% highest posterior density intervals (lower triangle).

credible band for the rate of change (Fig. 6, grey band) was wide, the majority of the rates of change for individual Monte Carlo iterations (Fig. 6, black lines) had the same shape, with a minimum in the middle (between depths 1 m and 3 m). The overall pattern of rate of change makes intuitive sense, given that on the centred logratio scale, all taxa had substantial changes in posterior mean predicted relative abundance near the surface, all but algae (Fig. 3D, inset) and *A. aurita* (Fig. 3H, inset) had flatter relationships at mid depths, and all but *D. cincta* (Fig. 3F, inset) had substantial changes near the bottom. This pattern is even easier to understand in ilr coordinates (Fig. S6). In a biologically meaningful basis (Section S6), coordinates representing the contrasts between algae and animals, *A. aurita* and *D. cincta*, *M. edulis* and other filter-feeders, and sponges and bryozoans and ascidians had approximately linear relationships with depth (Figs. 3B, 3D, 3E, 3F respectively). Coordinates representing the contrasts between bare wall and macroscopic organisms,

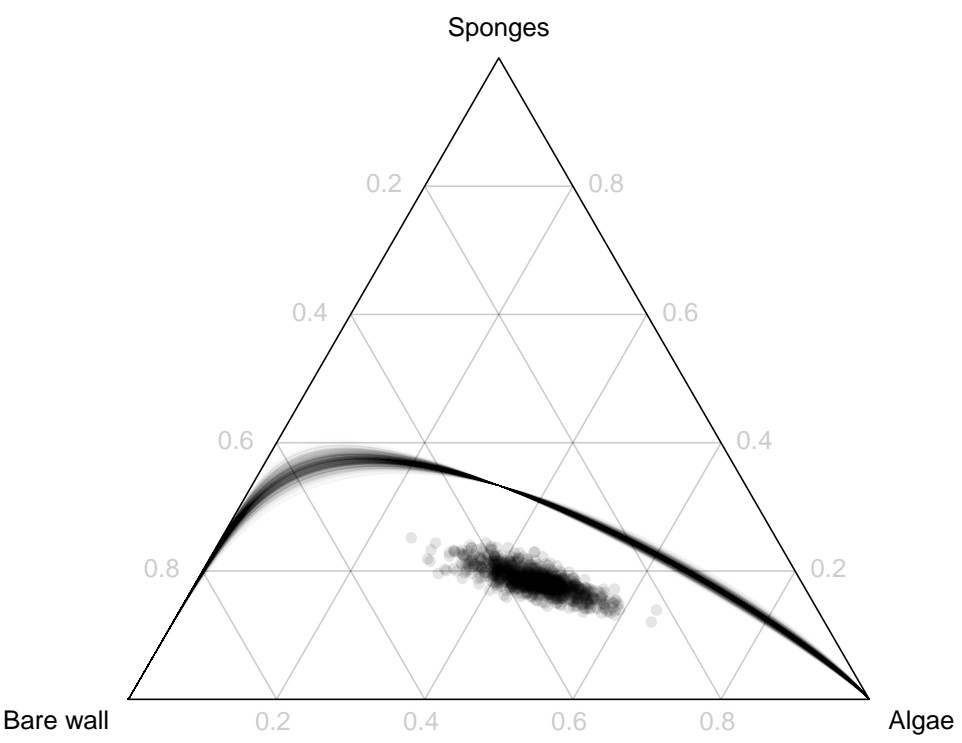

**Figure 5** The set of powerings of the depth coefficient $\gamma_1$ (lines, sample of 1,000 Monte Carlo iterations), and the squared depth coefficient $\gamma_2$ (dots: sample of 1,000 Monte Carlo iterations), for the subcomposition consisting of bare wall, sponges and algae.

predatory and filter-feeding animals, bryozoans and ascidians, and solitary and colonial ascidians had relationships with depth in which there was a clear minimum rate of change near the middle of the depth range (Figs. 3A, 3C, 3G, 3H respectively). Thus overall, the rate of change of location in ilr coordinates (and thus the rate of change of composition) was fastest in the middle of the depth range.

Using Bray–Curtis dissimilarity between adjacent predicted compositions led to a very different pattern of rate of change (Fig. S8), with local maxima at approximately 0.5 m and at 3 m. In the compositional data analysis framework, these local maxima would be seen as artefacts resulting from an inappropriate measure of compositional difference. Similarly, for the mite data, Bray–Curtis dissimilarities led to artefactual patterns in rate of change of community composition with respect to both water content and substrate density (Fig. S12).

## Mean composition of organisms over the entire depth

Over the entire depth range, bare wall had the highest relative abundance of around 0.5 (Fig. 7). This means that over half the area of the dock walls was not covered by any macroscopic organism. The macroscopic taxa with the highest relative abundances were sponges and solitary ascidians, with relative abundance around 0.2, followed by *Bugula*, with relative abundance around 0.05. These taxa, especially *Bugula*, did not have very

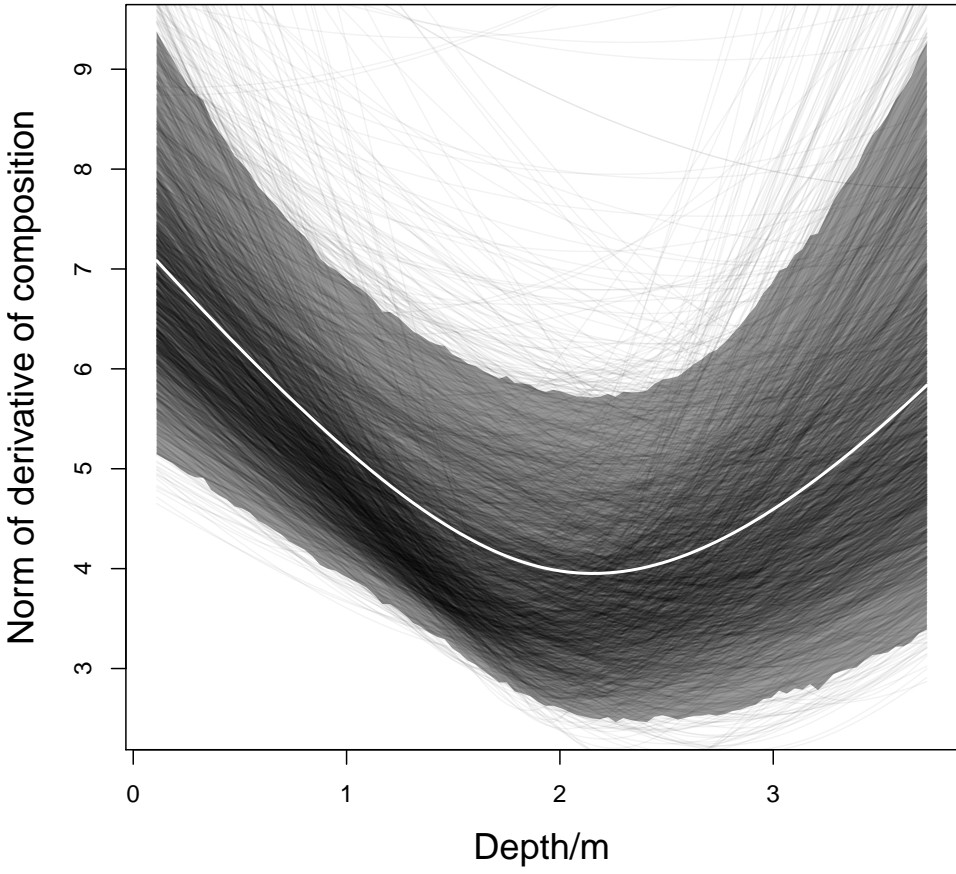

**Figure 6 Relationship between rate of change of community composition with respect to depth (the norm of the derivative with respect to depth) and depth.** White line: posterior mean. Grey band: 95% HPD credible band. Black lines: norms of derivatives for a subsample of 2,000 Monte Carlo iterations.

high relative abundance at any depth (Figs. 3B–3C, 3E), but had moderately high relative abundance at all depths, resulting in fairly high mean relative abundances. All other taxa had low mean relative abundances, including algae, which was very abundant at the surface but decreased quickly with depth (Fig. 3D).

## DISCUSSION

We showed that the vector space structure of the simplex leads naturally to tangible, functional and intuitive summaries of the changes in community compositions with depth in a subtidal marine system. A relatively simple quadratic model was a plausible description of these changes. This is important because needing a complicated model to describe observations is often a sign of some fundamental misspecification. For example, one reason to think that the Lotka–Volterra equations are generally useful is that they can be derived as a second-order Taylor polynomial approximation (*Lotka, 1956*, pp. 65, 78). Although a regression analysis cannot reveal the causes of the pattern we observed, it can hint at possible explanations. For example, integrating the composition over depth showed that

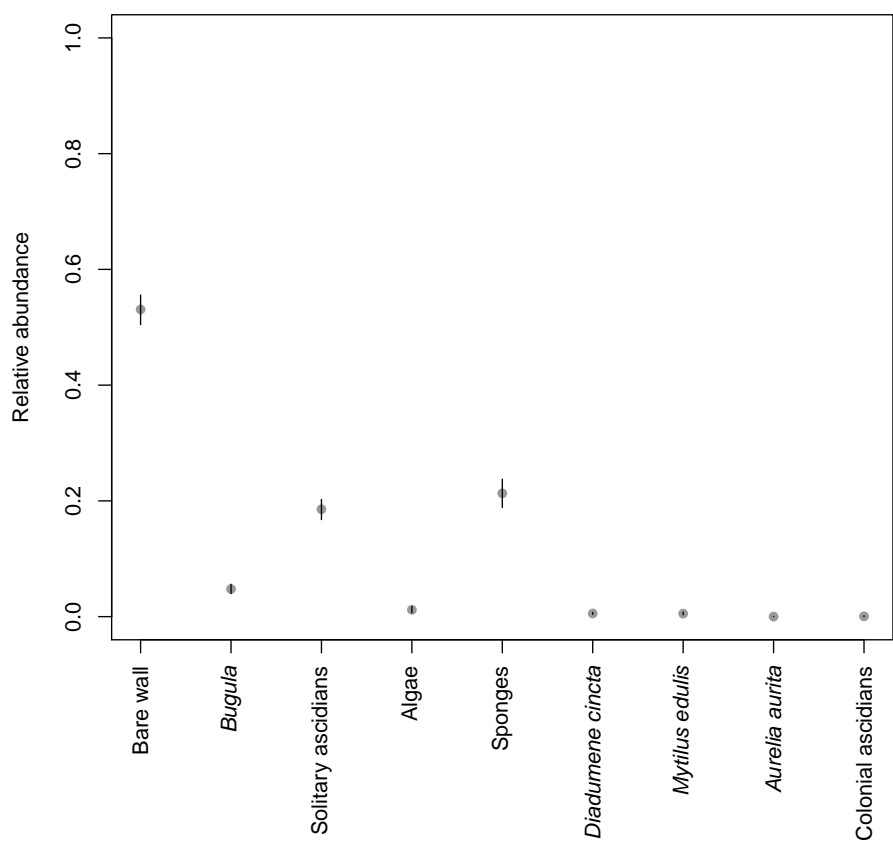

**Figure 7 Mean relative abundance of the eight taxa and bare wall, obtained by integration over the entire depth range.** Dots: posterior means. Black lines: 95% HPD intervals.

bare wall had much higher relative abundance than any taxon, suggesting that the classical picture of intense competition for space determining the structure of subtidal marine communities may need revision (*Ferguson, White & Marshall, 2013*; *Svensson & Marshall, 2015*). A major strength of the compositional data approach is the logical connection between statistical modelling and ecology. For example, we showed that the community was changing fastest at the surface and near the bottom, and that we would not find the same community composition at different depths. These results were based on a measure of dissimilarity that has both a strong statistical justification, based on the requirement for perturbation invariance (*Aitchison, 1992*) and a natural biological interpretation as the amount of among-taxon variability in proportional population growth rates needed to transform one community into another. In contrast, the popular Bray–Curtis dissimilarity, which is not perturbation invariant and does not have a natural biological interpretation, led to very different results. We therefore believe that compositional data analysis deserves to be more widely used by ecologists.

An observational study alone cannot determine the causes of the patterns in relative abundance with depth in our data. Although space is thought to be a limiting resource in many hard-substrate subtidal communities (*Witman & Dayton, 2001*, p. 356), it seems
unlikely that space is limiting at our study site, because of the high relative abundance of bare wall (Fig. 7). It is possible that bare wall is not available space after all due to the presence of biofilms that inhibit settlement. However, facilitative effects of biofilms on settlement are much more common in the literature than inhibitory effects (*Wieczorek & Todd, 1998*). It is also sometimes the case that apparently empty space is the result of intense competition between anemone clones. However, anemones were not abundant at our site, and the species we found do not have acrorhagi, the specialized tentacles used to deter other clones (*Hayward & Ryland, 1995*). Our surveys were done in winter, but relative abundance of bare wall remained high in summer (*Edney, 2017*), so it is unlikely that space is even seasonally limiting. Also, competition for space alone cannot explain the change in community composition with depth. Three other factors that may contribute to the depth effect are recruitment, food and oxygen availability.

Recruitment may regulate population dynamics of sessile marine organisms (*Caley et al., 1996*). For example, in a simple model for the dynamics of open populations of the bryozoan *Cellepora pumicosa*, equilibrium population size was proportional to recruitment rate (*Hughes, 1990*). At our site, settlement panels at 3 m typically had fewer than half as many new organisms as those at 1 m after five weeks in summer (*Edney, 2017*). Thus, changes in recruitment with depth are likely to contribute to the depth effect on community composition.

Competition for food may also be important. Experimental increase of phytoplankton supply increased species richness and reduced free space on settlement panels (*Svensson & Marshall, 2015*). Field measurements showed reduced phytoplankton density close to the walls in Albert Dock, the dock adjacent to our site (*Fielding, 1997*, p. 118). Thus, phytoplankton abundance may be limiting. However, it is not clear whether light levels will decrease with depth rapidly enough to generate a strong depth effect on phytoplankton production, and thus for phytoplankton limitation to generate a depth effect on community composition. For example, chlorophyll *a* concentrations in the Liverpool docks were little different between surface and bottom water (*Fielding, 1997*, p. 106).

Oxygen depletion may occur in the low-flow, topographically complex environment typical of fouling communities (*Ferguson, White & Marshall, 2013*). Summer oxygen levels in the Liverpool docks may be much lower near the bottom than the surface (*Fielding, 1997*, pp. 74–75). Thus exploitative competition for oxygen may become more intense as depth increases, potentially contributing to the depth effect on community composition, at least in summer.

The compositional regression approach taken here is closely related to multinomial logistic regression, but offers some advantages in flexibility and interpretability. Multinomial logistic regression is another approach to the analysis of count data derived from an underlying continuous model for relative abundances on a gradient (e.g., *Ter Braak & Van Dam, 1989*; *Qian, Cuffney & McMahon, 2012*). In multinomial logistic regression, the linear predictor is expressed in terms of logs of ratios of relative abundances, exactly as in a compositional linear model. In its basic form, multinomial logistic regression does not allow for overdispersion, which in a compositional linear model such as Eq. (1) is captured by the random intercepts $\varepsilon_i$ (*Xia et al., 2013*). Overdispersion is important for

describing aspects of sampling and biology that depart from the multinomial assumption, including variation in sampled area, clustering of individuals, as in the cnidarian *A. aurita*, and spatial extension of colonies, as in sponges.

More importantly, treating the simplex as a Euclidean vector space with perturbation and powering operations makes it easy to do algebra and analysis on compositions. This can simplify interpretation compared to the multinomial regression approach, where coefficients are expressed on the log-odds scale (*Billheimer, Guttorp & Fagan, 2001*). For example, we were able to determine why, in algebraic terms, we did not see communities with high similarity at widely separated depths, even though such an outcome is possible under a quadratic model. Such outcomes are related to the "double-zero problem" in the design of measures of ecological dissimilarity (*Legendre & Legendre, 2012* p. 271). A given taxon may have low expected relative abundance at both ends of a gradient because of unsuitable conditions. In our data, this pattern occurred for taxa including solitary ascidians and sponges (Figs. 3C and 3E). With finite sampling effort, this may lead to zeros at both ends of the gradient. However, unless the quadratic coefficient is an exact powering of the linear coefficient, the predicted dissimilarity will not be exactly zero. We therefore do not think that similarity resulting from similar relative abundance patterns is ecologically misleading, even if it does not arise from similar environments.

The algebra of perturbation and powering is central to visualization and interpretation of experiments and observational studies on compositional response variables. For example, *Billheimer, Guttorp & Fagan (2001)* expressed the effects of vegetation removal and addition of specialist predators on arthropod community composition, relative to a control treatment, using a perturbation. Similarly, *Billheimer et al. (1997)* used a perturbation to visualize the effect of salinity on relative abundances of stress-tolerant taxa, intolerant taxa and palp worms in a benthic habitat. In a regression study, *Xia et al. (2013)* visualized the estimated effects of changes in nine different nutrients on the relative abundances of three bacterial genera in the human gut microbiome as compositional straight lines, using the perturbation and powering operators. In all these cases, the necessary algebra is very straightforward if the simplex is treated as a vector space. Less obviously, knowing that a statistic has the perturbation invariance property (*Aitchison, 1992*) guarantees that differences in detection probabilities among taxa will not affect the results. For example, because we used the perturbation-invariant Aitchison distance as a measure of dissimilarity, our estimates of rate of change will not be biased by large, conspicuous organisms such as the solitary ascidians *C. intestinalis* and *S. clava* being easier to detect than small, inconspicuous organisms such as the cnidarian *A. aurita*. In contrast, widely-used dissimilarity measures such as Bray–Curtis, which is not perturbation invariant, would lead to artefacts.

## CONCLUSIONS

In conclusion, we believe that ecologists working with relative abundance data would benefit from making more use of compositional data analysis. There has been substantial progress in compositional data analysis since the 1980s, but as yet, it has had little influence on ecology. In areas such as the analysis of environmental gradients, compositional

data analysis provides a simple, coherent approach that is in keeping with the current preference for model-based analyses. With only a small shift in perspective, techniques such as differentiation and integration can be used to answer ecological questions in ways that have meaning for relative abundances.

## ACKNOWLEDGEMENTS

We are grateful to the 2017 ENVS271 class for ROV piloting, and to Juan José Egozcue, Cajo ter Braak and an anonymous reviewer for detailed and constructive comments on the manuscript.

### Funding

This work was funded by NERC grant NE/K00297X/1, and the University of Liverpool's Herdman Endowment. The funders had no role in study design, data collection and analysis, decision to publish, or preparation of the manuscript.

### Grant Disclosures

The following grant information was disclosed by the authors:
NERC: NE/K00297X/1.
University of Liverpool's Herdman Endowment.

### Competing Interests

The authors declare there are no competing interests.

### Author Contributions

- Fiona Chong and Matthew Spencer conceived and designed the experiments, performed the experiments, analyzed the data, contributed reagents/materials/analysis tools, prepared figures and/or tables, authored or reviewed drafts of the paper, approved the final draft.

### Data Availability

The raw data is available as Supplemental Files and on figshare: Spencer, Matthew (2018): Analysis of relative abundances with zeros on environmental gradients: a multinomial regression model. figshare. Media. https://doi.org/10.6084/m9.figshare.6851081.v1.

### Supplemental Information

Supplemental information for this article can be found online at http://dx.doi.org/10.7717/peerj.5643#supplemental-information.

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
