# Peer review of "Analysis of relative abundances with zeros on environmental gradients: a multinomial regression model"

_PeerJ, doi:10.7717/peerj.5643_

## Round 0.1 · original submission · Major Revisions

A 3rd reviewer could not submit due to time limitations, but I have been happy to also provide a review of my own. Below are a lot of suggestions. Following all of them goes probably beyond your time limitations, but I suggest to digest the most important ones (and my preference is visible below). I find it important that the paper reaches publication-readiness.

There are a number of aspects of this paper that could lead to a more interesting/informative paper. Personally I was more impressed by the supplementary material than by the paper, which is not a good sign. On first reading the paper appears to lack novelty, but on later readings, and helped by reviewer 1, more novelty comes about. Please make it more explicit. Reviewer 2 writes that ecologists who understand the intro probably use log-ratio analysis already and others will not be convinced. The reviewer suggests good simplifications. Use them and move the more technical stuff to later.

Broaden the definition of compositional data to be in agreement with line 31 (see also reviewer 1).

Please state more clearly which biological question is at stake and how the analysis contributed to answering the question. You could also speculate (or preferably demonstrate) how that differs from methods not satisfying the compositional assumption.

Please be more explicit on the treatment of zero abundance. How you dealt with zeroes (and how many there are) is hidden somewhere. For that reason, the second reviewer asked which pseudo-count was used. The treatment of zeroes is hidden in line 196. But is this line correct/complete? Your model is a glmm, so a multinomial with covariances on the link scale (is it?), and the link is the ilr (is it?).

You phrase your method in the context of log-ratio analysis. This analysis is based on the multivariate normal distribution and requires taking the logarithm of the data. The main obstacle for its use in ecological data is the multitude of zeroes. To try and circumvent this problem often a pseudo-count (often 1) is added, amounting to a clr or ilr of y+1. The pseudo-count can have a huge effect on rare species. Your method is a GLMM method, so that it does not require a pseudo-count. I suggest to change the title so that it contains the word ‘zero/zeroes’.

And the main novelty of the paper might be that you link the log-ratio world with the GLMM/multinomial logit/Goodman RC model world, most prominently being the symmetric multinomial model (eq (6.5) in http://data.princeton.edu/wws509/notes/c6.pdf, with eq (6.6) on link-scale) used in Friedman’s glmnet in R. For this, in ecological context, see (Jamil and ter Braak 2013) and references therein [replace the logit link of the paper by the log link to make it a real Goodman RC model for count data instead of 1/0 data]. Can the ilr be interpreted as a link function? What is the relation to and/or the advantage of your method over log-linear GLMM models with free row and column parameters (as in (ter Braak et al. 2017)). How do you deal with (residual) covariance (Warton et al. 2015, Ovaskainen et al. 2017); which role does ilr play in this?

Note that the multinomial model lives already in the simplex and the standard form is the multinomial logit model (MLM), used in ecology by me - sorry, same problem as the first reviewer – for the first time in (ter Braak and van Dam 1989), also in (ter Braak et al. 1993) but the clearest is perhaps (ter Braak 1995). Please explain how your proposed method relates to this. In these papers, MLM did not improve upon much simpler methods based on weighted averaging (which is Canonical correspondence analysis (ter Braak 1986) with a single prediction, and thus related to correspondence analysis). See also Ihm’ model B for Gaussian ordination (Ihm and van Groenewoud 1984) explained in section 7.1 of the Canoco 2.1 manual http://edepot.wur.nl/248698 and which is related to Goodman’s RC model which is a compositional model for count also. See (Jamil and ter Braak 2013, ter Braak 2017) and (De Rooij and Heiser 2005, de Rooij 2007).

The second reviewer asks for comparison with widely used method(s). The first logical choice is log-ratio analysis assuming normality after transformation. This requires a pseudo-count/number; show how sensitive this choice is by varying it. The second logical choice is the MLM model as in the above reference, e.g. using glmnet. How do these two methods compare to yours in terms of results and biological conclusions?

The above two models are already geared to compositional data. In the model-based world, of similar complexity as yours, we have (Warton et al. 2015, Ovaskainen et al. 2017). But Ovaskainen and Warton neglect the composional aspect or the fact that species may have niches, resulting in unimodal models. What happens if one bluntly applies their models, even when the row-sums are made equal?

In terms of Legendre & Legendre your method does Canonical analysis (aka constrained ordination). Another optional choice is the super-super simple model of (partial) canonical correspondence analysis; its relation to compositional data is given in Table 1 and the discussion of (ter Braak 1988). With a single predictor one does not have dimension reduction in the constrained part (it is simple Weighted averaging as first component, see above), but one still has residual covariance. In the ordination world the second and higher component represent residual covariance; these are unconstrained correspondence analysis axes that are orthogonal to the predictor. The method gives only an ordering of species along depth and an importance value (% explained) per species; no curves; no uncertainty. An optional addition only.

[Unconstrained ordination would be in the normal case result in (Aitchison 1983), a pca on clr data (with pseudo-count!); what would be your version? I do not expect this to be add to the current paper, but I mention it to set the context of Legendre & Legendre].

The second reviewer asks to include an analysis of kind of standard data set, with discussion on which biological conclusions might differ between methods. I support this wish. The reviewer suggests a 16S data set or one from Legendre & Legendre. Another (simpler) option is to use the data set dune (with dune.env) in the R-library vegan which is the example data set in (Jongman et al. 1995) [Almost identical to: http://edepot.wur.nl/410376]. You might select moisture as explaining variable if you must/want to select a single important variable.
Explain the feasibility of the method when there are not 8 categories but of the order of 100 or 1000 taxa (as in 16S data).
Relevant comparison of correspondence analysis and standard (multivariate normal - based) log-ratio analysis is in (Greenacre and Lewi 2009, Greenacre 2010). Most importantly, correspondence analysis does not adhere to subcompositional coherence. So, explain why this important for answering your question. Thus, expand on line 39. For ecological data like 16S the level of analysis is not fixed (order/family/genus) and distributional equivalence might be important. See Greenacre’s papers above.
For a discussion on Bray-Curtis, see the recent paper on size and shape by Greeacre in Methods in ecology and evolution.

Explain why you need or want to go Bayesian to fit model (I have nothing against it, but it brings a computational burden that must be warranted..).

I googled ilr and you then see that people are interested in knowing its (dis)advantages.

I suggest to move much of the intro in a first section (or as supplement) and replace it by easier material (the difficulties of compositional data) suggested by reviewer 2.

Reviewer 1 has a number of suggestions that are more or less easy to accommodate.



Reviewer remarks of editor

L25 How relates your work to this earlier work?

L 80 Different detection probabilities among taxa? But that problem is not solved by compositional analysis!

L214 Where is the dependence of the coefficient on the category/species? Is there none?? But see fig .

L214. The polynomial makes it a kind of old-fashioned curve fitting. As reviewer 1, are the square and third order terms needed? If it is about curve fitting anyway, make it modern, optionally change it to a Eilers & Marx P-spline, which easily fits in your framework and is easy to fit in your Bayesian context.

L244 norm of a difference = distance, is it not? Why do we need norm here?

L301 This is a rather late mentioning of zeroes!

L301 Relative abundance is zero or infinite. Delete ‘sample relative’ here.

Aitchison, J. 1983. Principal component analysis of compositional data. Biometrika 70: 57-65.
de Rooij, M. 2007. The distance perspective of generalized biadditive models: Scalings and transformations. Journal of Computational and Graphical Statistics 16:210-227.
De Rooij, M., and W. J. Heiser. 2005. Graphical representations and odds ratios in a distance-association model for the analysis of cross-classified data. Psychometrika 70:99-122.
Greenacre, M. 2010. Log-Ratio Analysis Is a Limiting Case of Correspondence Analysis. Mathematical Geosciences 42:129-134.
Greenacre, M., and P. Lewi. 2009. Distributional Equivalence and Subcompositional Coherence in the Analysis of Compositional Data, Contingency Tables and Ratio-Scale Measurements. Journal of Classification 26:29-54.
Ihm, P., and H. van Groenewoud. 1984. Correspondence analysis and Gaussian ordination. Compstat Lectures 3:5-60.
Jamil, T., and C. J. F. ter Braak. 2013. Generalized linear mixed models can detect unimodal species-environment relationships. PeerJ 1:e95.
Jongman, R. H. G., C. J. F. ter Braak, and O. F. R. van Tongeren. 1995. Data analysis in community and landscape ecology. Cambridge University Press, Cambridge.
Ovaskainen, O., G. Tikhonov, A. Norberg, F. Guillaume Blanchet, L. Duan, D. Dunson, T. Roslin, and N. Abrego. 2017. How to make more out of community data? A conceptual framework and its implementation as models and software. Ecology Letters 20:561-576.
ter Braak, C. J. F. 1986. Canonical correspondence analysis: a new eigenvector technique for multivariate direct gradient analysis. Ecology 67:1167-1179.
ter Braak, C. J. F. 1988. Partial canonical correspondence analysis. Pages 551-558 in H. H. Bock, editor. Classification and related methods of data analysis. Elsevier Science Publishers B.V. (North-Holland) http://edepot.wur.nl/241165, Amsterdam.
ter Braak, C. J. F. 1995. Non-linear methods for multivariate statistical calibration and their use in palaeoecology: a comparison of inverse (k-nearest neighbours, partial least squares and weighted averaging partial least squares) and classical approaches. Chemometrics and Intelligent Laboratory Systems 28:165-180.
ter Braak, C. J. F. 2017. Fourth-corner correlation is a score test statistic in a log-linear trait–environment model that is useful in permutation testing. Environmental and Ecological Statistics 24:219-242.
ter Braak, C. J. F., S. Juggins, H. J. B. Birks, and H. Van der Voet. 1993. Weighted averaging partial least squares regression (WA-PLS): definition and comparison with other methods for species-environment calibration. Pages 525-560 in G. P. Patil and C. R. Rao, editors. Multivariate Environmental Statistics. North-Holland, Amsterdam.
ter Braak, C. J. F., P. Peres-Neto, and S. Dray. 2017. A critical issue in model-based inference for studying trait-based community assembly and a solution. PeerJ 5:e2885.
ter Braak, C. J. F., and H. van Dam. 1989. Inferring pH from diatoms: a comparison of old and new calibration methods. Hydrobiologia 178:209-223.
Warton, D. I., F. G. Blanchet, R. B. O’Hara, O. Ovaskainen, S. Taskinen, S. C. Walker, and F. K. C. Hui. 2015. So Many Variables: Joint Modeling in Community Ecology. Trends in Ecology & Evolution 30:766-779.

·

Basic reporting

The manuscript is well written and, in general, easy to read. However, there are some points that require a more clear explanation (see points below). Also references are adequate and relevant although they can be complemented for a more exhaustive bibliography (see points below). Figures are well presented and they are relevant. Raw data and R/Stan scripts are available.

Experimental design

The addressed research is essentially methodological and consists of introducing compositional data methods in the field of Ecology. In my opinion it is within the scope of the journal.

The main goal of the manuscript is to introduce the compositional (log-ratio) approach in environmental-ecological fields. Special mention deserves the introduction of compositional derivatives and their interpretation to extract conclusions form the model. Also the treatment of zero-counts through a regression model estimated in a Bayesian framework can be considered as a novelty in this context of modeling species composition.

The problem and data treated are of moderate interest but the methods applied to model the community evolution along the depth clearly justifies the publication of the manuscript.

Validity of the findings

Data provided by the authors allows replication of results. Their principal virtue is that they represent a quite frequent situation, a several species composition changing along a gradient in which abundant zero counts appear. The attained conclusions are sound apparently valid and well based on the fitted model.

Additional comments

First of all I liked the manuscript. The methodology is adequate to model the available data set in a successful way. However, I think that it can be improved in some directions described below. I consider that these possible improvements are not necessary for the publication but for gaining quality/clarity.

IMPROVEMENT LINES

Some concepts on the log-ratio approach to compositional data can be more carefully described and documented. Some points follow:

1. Page 6, lines 127-130.In environmental, health and ecological literature, there is a belief that the information contained in relative abundances and the so called absolute abundances is different. As samples are not exhaustive (in mass, in volume or size) the absolute abundances are also relative to that size or mass and they can be considered as proportional to the relative abundances. In compositional data analysis proportional sets of abundances are considered "compositionally" equivalent.

2. Equation of Aitchison distance (Section 2.4.3). The formula is wrong: a parenthesis with a square is lost. This equation can be extended to the expression in terms of the ilr-coordinates. This is, the square Aitchison distance is the ordinary Euclidean distance between the ilr coordinates. This expression is used afterwards but just commented (page 13, line 287; remember that the square norm is the square distance of a composition to the neutral element).

3. Page 10, lines 225-229. Here the authors present the clr-coefficients as a way of interpreting the model efficiently. Being this true, it seems that each clr-coefficient is related to a single component of the composition (taxon). The sentence "Thus a constant slope on the clr scale corresponds to constant proportional change in the relative abundance of a given taxon" contains this association taxon-clr coefficient, and attributes the detection of "proportional change" to clr scale. Any log-ratio coordinate has this property, particularly the ilr coordinates used to fit the model. The problem with the slope in the clr scale is that this slope changes with the subcomposition (set of taxa) analysed.

4. Page 10, line 209. In order to fit the model, abundances are blindly ilr transformed. Although this is acceptable, the opportunity of interpreting the ilr-coordinates is lost. The design of a sequential binary partition (SBP) (Egozcue & Pawlowsky-Glahn 2005, Pawlowsky-Glahn et al. 2015) produces a set of balances as ilr coordinates. These balances can be easily interpretable in terms of equilibria between groups of taxa. This can be more efficient than the interpretation in terms of clr coefficients commented in point 3. On the other hand, linear change on compositions straight lines in the ilr-coordinate space (the clr-space is identical to that of ilr-coordinates).

5. Definition of compositional data (Abstract line 7; page 2,line 19). The authors use the early definition based on the constant sum of a vector of positive components. Nowadays, compositional data are better described as equivalent classes of proportional vectors with positive components. This leads to the conclusion that any composition can be represented as a constant sum vector but this representation is neither necessary nor sufficient condition for using the log-ratio approach for compositional data (e.g. Pawlowsky-Glahn et al. 2015, Barceló-Vidal and Martín-Fernández 2016.)

6. Although references are adequate they can be completed in some aspects. Some suggestions for additional references, most of them authored by me (J.J.Egozcue):

The name "Aitchison Geometry" and also "metric variance" and "metric centre" were firstly introduced in

Pawlowsky-Glahn, V. and J. J. Egozcue: Geometric Approach to Statistical
Analysis on the Simplex, Stochastic Environmental Research and Risk Assessment,
15, 5, 384-398, 2001.

Agreeing that compositional methods have low impact within the ecological literature, consider this reference where compositional methods are used:

R. López-Flores, X. D. Quintana, A. M. Romaní, Ll. Bañeras, O. Ruiz-
Rueda, J. Compte, A. J. Green, J. J. Egozcue: A compositional analysis
approach to phytoplankton composition in coastal Mediterranean wetlands:
Influence of salinity and nutrient availability. Estuarine, Coastal and Shelf
Science, 136, 72-81, 2014.

Fitting compositional differential models (not Bayesian) have been studied in

Egozcue, J. J. and Jarauta-Bragulat, E.: Differential Models for Evolutionary
Compositions, Mathematical Geosciences, 46, 4, 381-410, 2014.

An updated summary of compositional methods (including differential equations) can be found in

V. Pawlowsky-Glahn, J. J. Egozcue and R. Tolosana-Delgado: Modeling
and Analysis of Compositional Data, John Wiley & Sons, Chichester, UK, 2015

7. The description of the Bayesian model used is passed to supplementary materials. In my opinion a brief reference on the observational model and its main assumptions (residuals in ilr-coordinates are normal) can be presented in the body of the paper.

Reviewer 2 ·

Basic reporting

In this manuscript, Fiona Chong and Matthew Spencer present a dataset that utilizes video imagery to survey taxonomic variation as a function of depth and demonstrate that a compositional data analysis (CoDA) that is based on the “algebra of compositions” can successfully be applied to this dataset. This manuscript is timely as there has been a good deal of recent interest in how compositional artifacts can distort analyses of ecological datasets. The algebraic notation in the manuscript is dense and while this is perhaps unavoidable in may make the paper somewhat inaccessible to the target audience of experimental ecologists.

1. Basic Reporting. The authors sufficiently motivate the importance of compositional considerations when working with ecological data. The discussion of the compositional data analysis aspects was generally well executed.

- It may be of benefit to ecologists reading this article to be clear about the limitations of compositional data. This primarily comes about by the fact that with such relative data one is unable to know if an increase in relative abundance is due to an increase in the population of that taxon, all other taxa held constant, the maintenance of that taxon’s population given a decrease in other taxa, or a decrease in the population of that taxon with other taxa decreasing to a greater extent than the apparently “growing” taxon. This discussion may be helpful near lines 287-289.

- The addition of discussion regarding the interpretability problems surrounding the ILR and the potential mathematical limitations of the CLR is warranted.

- More detail about the Monte Carlo Markov chain resampling scheme should be put in the main paper.

- Figure 2: The caption is currently lacking information on the pseudo-count used in the CLR transformation. The changes in the y-axis scales of the inset images can potentially be misleading. It is suggested that the authors either set the scales to be uniform across all panels or add a supplemental figure with such a view of the data.

- Figure 6: A zoomed inset image within this figure or a supplemental figure showing the mean relative abundance of rare taxa would be helpful in providing a visual component to the discussion of the 95% HPD intervals for these taxa. This is currently difficult to decipher because of the fixed scaling with the more abundant taxa.

- Figure 1: Which imaging system (the ROV’s camera or the GoPro) was used to capture these images? The figure also appears to blur to various extents across the different depths.

- Figure 3: If (a) and (b) are symmetric along the diagonal, then compositing the 2 figures into one along the diagonal may enable readers to better discriminate similar coordinates between the fine gradations in the two figures which is difficult to do currently. Figure 3 as currently presented is quite difficult to interpret.

- Figure 4: The caption may be missing information on the numerical bounds of each variable which may aid in its interpretation by readers less familiar with CoDA methods. The additional of tick marks along the axes may offer similar help.

- Supplemental Figures and Tables: In order to avoid potential confusion, it may be necessary to reiterate that the p - 1 (8) ILR coordinates have no simple interpretation in terms of the (8 taxa + 1 wall = 9) initial p coordinates.

Experimental design

2. Experimental Design.
The main thesis of the manuscript is that analyses based on “compositional algebra” can give important insights into ecological data. This thesis would be considerably strengthened if the authors could contrast their analysis with the kind of standard analyses used in the ecological literature. How does their analysis compare to a simple linear model fit directly comparing a polynomial based on depth to the number of counts? Does a PCA or MDS ordination yield imperfect separation between depths that the authors’ pipeline is better able to discern? Some direct comparison of the newly developed pipeline to standard techniques (which the authors claim are misapplied in the literature) would increase the potential impact of the manuscript. The authors assert that “Most of the popular measures of community dissimilarity are not perturbation invariant, and are therefore misleading.” While this detail is technically correct, their paper would be strengthened by a demonstration of an incorrect biological conclusion that was drawn from such a “misleading” procedure and that their method produces a correct interpretation.
1. Can the authors be more explicit about whether they used a pseudo-count? The discussion of pseudo-count information for the CLR transformation is currently missing (lines 225-229). This is especially important information as the CLR transformation is used on rare taxa that likely have 0 values, which would lead to a geometric mean denominator to be 0. Given the small number of different taxa, it may be also be worthwhile to assess the behavior of the models given different pseudo-count values. Having this information would be especially helpful in the interpretation of Figure 2. The authors also do not discuss pseudo-count for the Aitchison norm which is warranted given its reliance on the geometric mean like the CLR transformation.

2. The description of the data collection process was difficult to follow and may be much improved by the addition of a simple diagram visualizing the sample collection and image analysis steps taken. It is also not clear what steps in the identification of organisms from the video and images captured were through automated image analysis or human visual curation (lines 171-176).

3. The authors mention the constraint on the number of points sampled always being 100 per still image (lines 195-198), but they also state earlier that they have removed two rare taxa from the analysis (lines 186-187). Were these points resampled as was the case for points that could not yield positive identification or were there, in fact, a few samples that had less than 100 points?

Validity of the findings

Validity of the Findings

The authors investigated the depth-dependent changes in community structure using perturbation-invariant techniques from the treatment of the relative abundance as compositional data. They reiterate the suitability of CoDA techniques for ecological data and encourage its increased usage by ecologists in arguments that attempt to tie together mathematics and biology. They also offer several ecological hypotheses explaining the dependence of community structure with depth.

1.The difficulties encountered during the data collection process regarding the failure of the ROV’s distance finding capabilities (lines 157-160) bring up questions regarding the robustness of the data. The authors assert that problems involved in their data set are unlikely to introduced errors in the analysis (lines 160-161), but they need to provide evidence supporting this claim. In particular, the claim that “the field of view was always large enough to contain many organisms, so the relative abundances are unlikely to depend on the exact area sampled” seems poorly justified.
2.It may best serve the ecologists who can potentially benefit from using the methods outlined in this article if the authors also include a secondary “classic” dataset from a source such as Legendre and Legendre. Such a replication dataset would enable their audience to be able to readily compare and contrast this method of analysis to traditional ones.

3.The statement calling into question the “classical picture of intense competition for space determining the structure of subtidal marine communities” is speculative. There are numerous microbial biofilms that may render such sections of habitat (“blank wall”) unsuitable for the growth of organisms documented in this study. Furthermore, there is the classic example of the “no man’s land” that forms between two competing colonies of sea anemones which is apparent barrenness that is the result of an intense competition for space.

4. Per PeerJ’s policies, all scripts and raw data should be made publicly available, and the video recordings were not among the material provided. This seems particularly important in a paper such as this one which seeks to introduce a new method.

Additional comments

Optional: The impact of the paper might also be substantially improved if the authors could demonstrate that their techniques worked on sequence/count datasets since many ecologists use these kinds of sequencing experiments to survey microbial diversity. There is for example a 16S sequence dataset that has been published ( https://www.nature.com/articles/ismej200986 ) that surveys microbial diversity as a function of depth. Applying the proposed analytical framework to such a dataset would potentially be of interest to ecologist, but may be outside the scope of the paper so is here included as an optional note to the authors.

---

## Round 0.2 · Minor Revisions

As with the reviewers, I found the paper tremendously improved in clarity and positioning compared to rival methods. Please consider the very minor and some optional suggestions of the two reviewers for the final version.

I have two additional suggestions. On line 185 you write that there is no clear distinction between fixed and random effects in Bayesian context. I teach my students that fixed effects are effects with a fixed (often uninformative) prior, that is without unknown parameters, whereas random effects have priors with unknown parameters that must be estimated from the same data.

The second suggestion is to place the tables and figures in the Supporting information in text instead of at the end; you should know that the way you format the Supporting Information is the final published version; it is not reformatted by PeerJ.

·

Basic reporting

No comment

Experimental design

No comment.

Validity of the findings

No comment.

Additional comments

In page 17, line 283, Equation of Aitchison distance, last right-hand-member: there is a typo. The last x should be subscripted as x_{2,j} .

I feel that your paper is an excelent contribution to the compositional data analysis.

Reviewer 2 ·

Basic reporting

no comment

Experimental design

no comment

Validity of the findings

no comment

Additional comments

The manuscript is much improved in clarity over the previous version and I have no further requirements for revision.

The below notes are minor and should be considered as optional suggestions:

Bray-Curtis is a dissimilarity and not a true distance as it does not preserve the triangle inequality.

The authors may find the following reference helpful in regards to the discussion of the using the Aitchison distance in dissimilarity analyses on lines 82-85: Microbiome Datasets Are Compositional: And This Is Not Optional (Gloor et al., https://www.frontiersin.org/articles/10.3389/fmicb.2017.02224/full), as the authors of that work similarly emphasize said point.

While not necessary suggested references for this manuscript, the authors may find the following R/python packages insightful for future work in aiding the biological interpretation of ILR balances: A phylogenetic transform enhances analysis of compositional microbiota data by Silverman et al. (https://elifesciences.org/articles/21887) and Balance Trees Reveal Microbial Niche Differentiation by Morton et al. (https://msystems.asm.org/content/2/1/e00162-16)

The authors may also find this methodology regarding Bayesian methods in analyzing compositional microbiome data helpful in future work: Unifying the analysis of high-throughput sequencing datasets: characterizing RNA-seq, 16S rRNA gene sequencing and selective growth experiments by compositional data analysis by Fernandes et al. (https://microbiomejournal.biomedcentral.com/articles/10.1186/2049-2618-2-15)

---

## Round 0.3 · accepted · Accept

Your revision addresses all points and I am grateful you took care of them. The paper is now well-positioned and very readable and welcome.

#